# Revisiting Embeddings for Graph Neural Networks

**Skye Purchase**
University of Cambridge
atp45@cam.ac.uk

**Aaron Zhao**
University of Cambridge,Imperial College London
a.zhao@imperial.ac.uk

**Robert Mullins**
University of Cambridge
robert.mullins@cl.cam.ac.uk

## Abstract

Current graph representation learning techniques use Graph Neural Networks (GNNs) to extract features from dataset embeddings. In this work, we examine the quality of these embeddings and assess how changing them can affect the accuracy of GNNs. We explore different embedding extraction techniques for both images and texts; and find that *the performance of different GNN architectures is dependent on the embedding style used*. We see a prevalence of bag of words (BoW) embeddings and text classification tasks in available graph datasets. Given the impact embeddings has on GNN performance. this leads to a phenomenon that *GNNs being optimised for BoW vectors*.

## 1 Introduction

Current advancements in Graph Neural Networks (GNNs) are being evaluated on a small range of tasks and accompanying datasets. Though these datasets are sourced from different domains, they require preprocessing the raw data into a computationally digestable format to be usable by GNNs, referred to as *embeddings*. In this work we focus on node classification and thus *node embeddings*.

Common node classification datasets [2, 5, 9, 12] focus on text classification with the primary node embedding being Bag of Words (BoW). Though this is a suitable method for text, this results in current GNNs being optimised to BoW. Equally, this form of node embedding is not always applicable, image data for example, and so GNNs are only being optimised for limited forms of data, mainly text. Existing literature has focused on the shortcomings of GNN training and the effect that the dataset can have on the model performance [9], but there is no comment on **how different node data preprocessing may affect performance**.

To demonstrate this problem we introduce three new datasets as alterations of existing datasets that are commonly used in literature. Each dataset is accompanied by a set of node embeddings. To evaluate the effect of node embeddings on GNN performance we train and test standard GNN archictures: Graph Convolution Network [5], Graph Attention Network[11] (with GATv2 [1]), and GraphSAGE [2] with two different samplers. For these models we find that their performance and relative rank is dependant on the embeddings used. In this work we make the following contributions:

- We put forward three new datasets and a rich set of accompanying embeddings to better test the performance of GNNs.

- We demonstrate that GNN performance depends on the embedding used. The choice of embedding provides large variance and prevents a fair comparison of different architectures.

- We demonstrate that current GNN architecture design overfits to limited styles of embedding.

S. Purchase et al., Revisiting Embeddings for Graph Neural Networks (Extended Abstract). Presented at the First Learning on Graphs Conference (LoG 2022), Virtual Event, December 9–12, 2022.

## 2 Background and Related Work

Given a graph, $\mathcal{G}(\mathbb{V}, \mathcal{E}, \boldsymbol{X})$, with raw node data, $\boldsymbol{X}$, there exists a transformation function, $f_e$, to project the raw data to a more compact feature space, $\boldsymbol{X}_e$, such that $\boldsymbol{X}_e = f_e(\boldsymbol{X})$. A GNN then trains and is evaluated on this transformed node data rather than the original raw data. A *(dataset ($\mathcal{G}$), embedding ($f_e$))* pair is a specific graph, $\mathcal{G}$, with a specific embedding function, $f_e$.

This work focuses on the standard set of Graph Neural Networks (GNNs): Graph Convolution Network (GCN) [5], Graph Attention Network (GAT [11] and GATv2 [1]) and graphSAGE [2]. These proposed models were evaluated using a small selection of datasets including the citation networks, Reddit, Amazon and Flickr. These datasets are also prevalent in current literature as a method of comparing new GNN architectures against prior architectures. This results in any shortcomings in these datasets propagating through successive papers. To combat this Hu et al. [4] developed the Open Graph Benchmark to standardise the datasets used for comparison.

However, we find that the vast majority of datasets focus on text classification tasks and utilise bag of words (BoW) extensively. **This is not a good representation of all tasks that GNNs may be used for** as BoW is not always applicable to the raw data. Furthermore, simple text classification does not require particularly complex or rich node features and therefore does not test the capabilites of GNNs.

Shchur et al. [9] focused on common pitfalls in GNN evaluation. Changing the train, evaluation and test split on a dataset can cause large changes in accuracy and the rank of different GNN architectures, even though other hyperparameters remained constant. This paper supports the idea that there are many factors that might affect the evaluation of the performance of GNNs. However, Shchur et al. [9] did not look into the embedding of the data which we find to be important for model performance.

## 3 Datasets

We introduce three new datasets: Flickr_v2, to highlight the importance of embeddings for non-text databases, and two Amazon datasets (AmazonElectronics and AmazonInstruments), to evaluate new text embeddings against the standard bag of words (BoW) approach. In each dataset we carry out a meta-label approach to generate labels for each node. Each node in the three datasets has a set of tags or categories which are converted into a word vector through GLoVe. These vectors are compared to each meta-label vector and the closest is chosen as the new label. Review and comparison to other datasets is detailed in Appendix A.

### 3.1 Flickr_v2

This is an image classification task using the same graph structure as the prior Flickr [7] dataset. Each node represents an individual image in the network where the raw data is the image data itself. Each edge represents some connection between images based on comments, likes and groups. As stated before, **in the case of images, BoW is not applicable**. Instead using image classification modelss provides a method of encoding raw image data in a compact feature space. Extracting these node embeddings from this feature space yields a sensible embedding to be used in a graph dataset.

We can initialise these image classification models with pre-trained weights to embed the images rather than needing to train the model ourselves. This does also open up the possibility of carry out further training of the image classification model to be better suited to GNN providing adaptable node embeddings. For the Flickr_v2 we use a selection of convolutional neural networks (CNNs), namely two variations of ResNet [3] (ResNet18 and ResNet50) and VGG16 [10], to provide the three embeddings for the Flickr dataset.

It is important to highlight the exclusion of BoW from Flickr_v2 as this is the embedding used in the current Flickr dataset [7]. Given that Flickr_v2 uses raw image data we cannot sensibly generate BoW embeddings and instead need to use an image based technique. The previous Flickr used human descriptions of the images which is not always possible when all that is available is raw image data.

### 3.2 AmazonElectronics and AmazonInstruments

Both Amazon datasets are text classification tasks using the graph structure induced by the "similar items", "co-viewed" and "co-bought". Each node represents a single item from a the specified category where the raw data is the review text.

In comparison to Flickr_v2 BoW is a suitable candidate as an embedding as the raw data is text reviews. Keeping with the approach for Flickr_v2 we utilise a text transformer model, specifically the roBERTa [6] transformer. Compared to the CNNs we have multiple stages in the classification to extract embeddings from: the preprocessing step converting the raw text into byte-pair encodings, the transformer encodings and the final feature vector before classification. These three embeddings are called Byte-Pair, roBERTa-Encoded and roBERTa respectively.

The roBERTa model also uses pre-trained weights in this case we use a roBERTa model trained on the MNLI dataset. This does mean that this specific model is not optimised for simple text classification, as MNLI is designed for sentiment analysis.

## 4 Evaluation

### 4.1 Experimental Setup

Each GNN is setup following their respective papers, in all cases this entails a 2 layer architecture with a final classification layer. Each of these architectures remains the same across all (dataset, embedding) pairs. As the datasets are based on existing datasets where the graph structure and input vector sizes are similar we use the same hyperparameters as the original papers. To prevent bias from training we use the same optimizer.

Each dataset is split into train, validation and test splits into 70%, 10% and 20% respectively. The same split is used across all the embeddings for a given dataset to prevent this influencing the performance. Each epoch of training uses the train and validation splits with the test split held out for evaluation. Each GNN is given 300 epochs to train on a (dataset, embedding) pair and we carry out 3 runs to calculate a mean accuracy and confidence interval.

**Table 1:** Test accuracy on Flickr_v2 with different embeddings.

| Model | Embedding Styles | | |
|---|---|---|---|
| | ResNet18 | ResNet50 | VGG16 |
| GCN | $41.8\% \pm 0.4$ | $38.3\% \pm 0.5$ | $\mathbf{35.5\% \pm 0.3}$ |
| GAT | $38.1\% \pm 0.6$ | $37.1\% \pm 1.1$ | $27.3\% \pm 1.2$ |
| GAT2 | $42.1\% \pm 1.8$ | $41.0\% \pm 1.5$ | $34.2\% \pm 0.8$ |
| GraphSAGE (Random) | $\mathit{45.4\% \pm 0.1}$ | $\mathbf{47.0\% \pm 0.0}$ | $\mathit{35.2\% \pm 0.2}$ |
| GraphSAGE (Neighbour) | $\mathbf{45.8\% \pm 0.2}$ | $44.5\% \pm 0.1$ | $34.5\% \pm 0.2$ |

### 4.2 Flickr_v2 and AmazonElectronics Results

Table 1 demonstrates how the different image node embeddings affect the performance of the five models. We see in the case of ResNet the best performing model is graphSAGE with the only difference being the sampler. This is contrary to the previous results seen on the prior Flickr dataset where Graph Attention Network (GAT) and the improved version GATv2 out-performed GraphSAGE.

When looking at the results for VGG16 we notice that there is less variation in the results for each GNN and a reduction in the accuracy of the models. However, when trained on the underlying images VGG16 out-performed both ResNet18 and ResNet50 achieving 47.0% accuracy compared to 45.2% and 46.9% respectively for the ResNet models. Therefore it is not only the performance of the network used to create the embeddings that is important but rather the feature vectors produced before classification.

We also see that the ranking of the models remains relatively consistent across the embeddings, though in the case of VGG16 the models perform relatively the same. Importantly **this ranking is different from those presented when evaluating on the bag of words (BoW) version of Flickr**.

Table 2 demonstrates how different text node embeddings affects the performance of the models. It is clear that the common standard of Bag of Words (Bow) is far superior to other embedding styles. What is more interesting is that fact that in this case we see that both GAT models out-perform the other models by a significant margin. But when looking at the roBERTa encoding we see that GraphSAGE performs the best in line with what we see in Flickr_v2.

**Table 2:** Test accuracy on AmazonElectronics with different embeddings.

| Model | Embedding styles | | | |
|---|---|---|---|---|
| | Bag of Words | Byte Pair | roBERTa Encoded | roBERTa |
| GCN | 69.1% ± 0.1 | 21.7% ± 0.2 | 22.7% ± 1.1 | 22.3% ± 1.2 |
| GAT | *81.1% ± 0.2* | 22.2% ± 0.5 | 46.1% ± 1.5 | 40.3% ± 2.9 |
| GAT2 | **81.8% ± 0.3** | 22.2% ± 0.6 | 41.8% ± 5.1 | 35.7% ± 5.6 |
| GraphSAGE (Random) | 71.3% ± 0.1 | *26.3% ± 0.3* | *57.0% ± 0.5* | *53.7% ± 0.5* |
| GraphSAGE (Neighbour) | 76.4% ± 0.3 | **40.4%** ± 0.4 | **67.8%** ± 0.4 | **66.4%** ± 0.3 |

Unlike in the case of Flickr_v2 we see that the results are more inline with the results we see from previous BoW datasets. We also see that the ranking of the models follows those that are presented in prior papers suggesting that **improvement of models has been focusing on optimising for BoW embeddings**. In this case the best embedding option is BoW, however, not always applicable.

It is important to note that we are not promoting an alternative embedding function other than BoW, since BoW shows the best performance in Table 2. However, we are showing the phenomenon that **current GNN architecture design and evaluation is promoting overfitting to BoW embeddings**. We see in the majority of node embeddings, which are not BoW, that GraphSAGE performs the best. This suggests that when using a wider range of embeddings our expected accuracy is better if we were to use GraphSAGE over GAT even though GAT is often consider state of the art. Results for AmazonInstruments are available in Appendix B.

### 4.3 Discussion and Limitations

**Discussion** In the case of text classification where our labels are representations of words or phrases we anticipate BoW to perform best. This is because BoW provides a discrete collection of word presence, if these can be linked to words in the labels then there is a simple direct connection between a few bits in the node vector and the output classification. In the case of Amazon the BoW vectors contain direct synonyms of the label words or phrases.

This invites the question as to why GAT performs far better on BoW whereas GraphSAGE performs better on model embeddings. In the case of BoW we have discrete inclusion or exclusion of a specific word but in the case of model embeddings we have a continuous vector that varies within the model's feature space. Therefore, it is more likely that GAT is good at picking out discrete features than GraphSAGE, which is to be expected given the architecture can utilise multiple heads to focus on individual entries. On the contrary GraphSAGE is better suited to continuous vectors as it takes the whole vector into account at once when computing a classification.

In comparison, the field of NLP is moving away from BoW. This is mainly because more complex tasks such as sentimental analysis or language modelling benefit from richer embeddings This in turn questions the focus on text-based graph datasets: **is text classification on a graph really the task we want to use to assess the quality of our GNNs?**

**Limitations** The datasets we provide are a small subsection of all possible representation learning tasks that could be carried out on graph networks. Similarly we only provide a handful of embeddings and do not endeavour to find the optimal embedding for each tasks. Thus these results do not represent all possible GNN tasks however we do still see clear trends. Due to the brevity of an extended abstract we only focus on variations on three standard models rather than analysing the effect of embeddings on more niche models. There has been a lot of work into simplified GNNs moving away from the layered approach of the models presented in this paper. These new models may be better suited or more consistent across the different node embeddings.

## 5  Conclusion

Current approaches to evaluating graph neural networks (GNNs) focuses on text classification using bag of words (BoW) embedding to transform the raw text into a compact node feature. However, this approach is not general for all types of data and thus the evaluation of GNNs is overfitting to BoW and text classification.

Our work demonstrates how the GNN performance is dependent on the node embeddings used in training and evalutation, providing new embedding candidates where BoW is not applicable. In evaluating different choices of embeddings we introduce three new datasets each with their own set of embeddings. We show that each node embedding favours different GNN architectures rather than simply effecting the accuracy.

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

# A    Comparison to Other Datasets

We aim to maintain the same graph structure as previous versions of our three datasets describted below. This is to keep the number of varying factors to a minimum and just focus on how the embeddings effect the result. To achieve this we use the same adjacency matrices published with these prior datasets. Although we were unable to source all the raw data these deviations are limited.

Due to the importance of $f_e$ on GNN performance there is a lot to discuss about prior datasets that exist within the space of GNNs in regards to these functions.

## A.1    Pytorch Geometric

Pytorch Geometric python library provides a standard interface on top of Pytorch to allow for the development of graph based machine learning. The library also provides a sample of datasets from previous papers published in this field.

As is clear from the table the current standard embedding for datasets is bag of words. In the cases where bag of words approaches are not used the approach is grounded in classical text representations such as n-grams and word vectors.

The tasks in these popular datasets are node classification where the node data is frequently text. We therefore say that on the node level these are text classification tasks. The only instance of a non-text classification task is Flickr [12], though based on the fact that the underlying data is image descriptions this could also be considered a text classification task.

This demonstrates how limited the reach of GNNs currently stand as they are being trained on datasets that behave very similarly where the only difference is the specifics of the available data. We feel that this does not therefore fully test the capabilities of GNNs and puts too much emphasis on bag of words and text classification.

## A.2    Open Graph Benchmark

The results in this paper focus on *node property prediction* as the data that unconnected models ordinarily work on is easily transferred to nodes in a graph. So when discussing Open Graph Benchmark [4] the focus is on the node property prediction subset (OGBN).

The goal of OGB is to create a standard set of datasets that can be used to compare different GNN architectures so a discussion as to way we did not use their datasets is warranted. The available datasets *ogbn-products, ogbn-proteins, ogbn-arxiv, ogbn-papers100M* and *ogbn-mag* all use variations on the same text representations used in Appendix A.1. These include Bag of Words (BoW), word2vec and skip-gram. This means the same discussions on these classical text holds here.

We see that the majority of the tasks focus on text classification, excluding *ogbn-protein*, this again draws into question how well these datasets are testing the range of classification tasks. Further to this, focusing mainly on BoW style embeddings raises the question of whether we are building good BoW extractors or graph information extractors.

## A.3    Flickr

The prior Flickr dataset used in *Zeng et al.* [12] originated from *McAuley et al.* [7] which aimed to utilize network connections and image descriptions rather than the images themselves. The specific embedding function that the paper used is Bag of Words.

This embedding function is a valid representation of images but it is not easily applicable to other image datasets. Thus GNNs trained on this dataset are confined to images with descriptions that have been transformed using the same top 500 words. Noting that this list of top 500 words is not readily available.

## A.4    Amazon

*Zeng et al.* [12] also provide an Amazon dataset (AmazonProducts) covering the entirety of Amazon. Without a known source we instead use available Amazon databases online to download and generate our own dataset. The embedding function used is to tokenise the reviews by 4-grams and take the

single value decomposition. This is, as with Flickr, not easily applicable outside of the original dataset.

An alternative Amazon dataset (Amazon) is also available from *Shchur et al.* [9] created originally in *McAuley et al.* [8]. Though the original source of the dataset used a pre-trained Caffe model to embed the product images this dataset did not use these. Instead they created their own embeddings using the bag of words standard with the product reviews as the raw data.

## B   Further Results

Table 3 contains the further results collected for AmazonInstruments

**Table 3:** Test accuracy on AmazonInstruments with different embeddings demonstrating how the different embeddings effects the performance and relative ranking of GNN models. Included is the standard deviation of each result.

| Model | Embedding Styles | | | |
|---|---|---|---|---|
| | Bag of Words | Byte Pair | roBERTa Encoded | roBERTa |
| GCN | $64.0\% \pm 0.5$ | $20.8\% \pm 0.3$ | $20.4\% \pm 0.8$ | $20.4\% \pm 0.8$ |
| GAT | *79.3% ± 0.6* | $21.6\% \pm 0.9$ | $47.5\% \pm 1.9$ | $46.1\% \pm 4.3$ |
| GAT2 | **79.4% ± 0.3** | $21.2\% \pm 0.6$ | *49.8% ± 5.0* | *47.8% ± 2.8* |
| GraphSAGE (Random) | $67.5\% \pm 0.3$ | *23.9% ± 0.6* | $45.1\% \pm 1.2$ | $41.9\% \pm 0.6$ |
| GraphSAGE (Neighbour) | $72.6\% \pm 0.3$ | **43.4% ± 0.5** | **62.4% ± 0.5** | **59.9% ± 0.6** |

## C   Hyperparameters

Table 4 details the layers of each model used providing the output hidden features of each layer, the sampler used (the specifics shown in Table 5) and the maximum and minimum learning rates. Where there is a difference in learning rates we use a learning rate scheduler that decreases the learning rate when validation accuracy plateaus. Where two models use the same sampler the parameters of those samplers are identical to keep consistency across the tests.

For GraphSAINTSampler all setups use a walk length of 2 with 5 steps sampling 100 nodes per node for normalisation calculation.

**Table 4:** Model architecture, sampler and learning rate

| Model | Hidden Features | Sampler | Learning Rate | |
|---|---|---|---|---|
| | | | Max. | Min. |
| GCN | 256 256 | Random Node | 1e-2 | 1e-2 |
| GAT | 256 256 | GraphSAINT RW | 1e-2 | 1e-2 |
| GAT2 | 256 256 | GraphSAINT RW | 1e-2 | 1e-2 |
| GraphSAGE (Random) | 256 256 | Random Node | 1e-3 | 1e-3 |
| GraphSAGE (Neighbour) | 256 256 | Neighbour | 1e-3 | 1e-3 |
| ResNet18 | *as provided* | - | 1e-4 | 5e-6 |
| ResNet50 | *as provided* | - | 1e-4 | 5e-6 |
| VGG16 | *as provided* | - | 1e-4 | 5e-6 |

**Table 5:** Sampler parameters

| Sampler | Dataset Split | Setup |
|---|---|---|
| GraphSAINT RW [12] | Train | roots: 6000 |
| | Validation | roots: 1250 |
| | Test | roots: 2000 |
| Random Node | Train | # partitions:512 |
| | Validation | # partitions:128 |
| | Test | # partitions:256 |
| Neighbour [2] | Train | # neighbours:[25, 10], batch size:512 |
| | Validation | # neighbours:[25, 10], batch size:128 |
| | Test | # neighbours:[25, 10], batch size:256 |

