# OpenReview forum: "Revisiting Embeddings for Graph Neural Networks"
_logconference.io/LOG/2022/Conference — LoG 2022 Poster_

### Official Review · Reviewer_e5xZ · 2022-09-30

**Overall Score:** 3
**Confidence:** 4

**Review:**

The paper writing definitely needs heavy rewriting. It is very hard to follow and I think it is not ready for publication.

This paper introduces a new perspective that GNNs can overfit certain feature types, and also GNN can help in some feature preprocess procedures. However, the paper is very hard to follow with many misleading words. I am still curious about what "the choice of embedding influences the selection of GNNs regardless of the underlying dataset" means after reading the whole paper.

I suggest giving more clear definition of the embedding and features. And the Graph-connected Network is somehow also misleading. I do not think there is much difference to view the message passing as the feature preprocess or the network structure. The SGC already bridge this gap.

On the experiment part, The most popular OGB dataset with raw text feature is not included into the discussion. The AmazonElectronics benchmark dataset is not a great dataseMoreover, it is no wonder that GNN does not work well in some situations. Whatever, the Bag of Words embedding performs the best across all models.

The paper use the wrong template I think, please renew to the latest one.

---

### Official Review · Reviewer_QyuE · 2022-10-14

**Overall Score:** 3
**Confidence:** 4

**Review:**

This paper studies the effect of feature encoder on the performance of graph neural networks. It proposes a model that fine-tunes the feature encoder model taking graph structure into account.

My recommendation is clear reject. The paper is REALLY hard to follow, it was probably the single hardest paper to parse and understand what is actually going on. The submitted version spends 2 pages for introduction, and half a page to actually describe the method, entirely omitting or referencing to appendix for even broad description of the proposed technique. The experimental results are weak and not supportive of the point that the paper is trying to make.

If I may, I would like to suggest a change in terminology to make the paper clearer. "Revisiting embeddings for GNNs" suggests positional encodings for nodes, and (L3) "extract features from dataset embeddings" suggests that features are somehow extracted from the embeddings of the dataset. I would like to suggest to use the terms "feature encoders" to denote that the paper concerns itself with representations of nodes' features.

To highlight some positive aspects of the work, I believe that the problem is actually very interesting. It is indeed an open question -- how to best integrate feature encoders and graph neural networks for multimodal datasets? It is unfortunate that this work does not provide a coherent answer to that question. I am hopeful that future iterations will be able to answer that question.

As far as I could understand it, the paper attempts to make the following observations (I'm following contributions from L41):
1. New datasets (Flickr_v2) and different featurization of these embeddings. However, these featurizations are mostly useless, as evident from Tables 2 and 3. In case of text data, the paper does not provide a representation that beats simple bag-of-words, which is standard in the field. Furthermore, other encoding variants significantly degrade model performance - it is unclear why anyone would prefer to use such models, as they are clearly not suitable for the task.

2. Some GNN models do not improve upon the baseline using the feature embeddings with no graph information. However, it is only the case with esoteric encodings introduced in the paper or the new dataset that is introduced there. The claim that this happens in standard datasets is not thus supported. There are datasets where graph data contains little, if at all, signal. Using graph-less methods as baselines is a standard practice when introducing datasets (e.g. see "pitfalls of GNN evaluations" or "benchmarking GNNs" papers).

3. A joint learning method, GraNet, improves upon feature embeddings in these scenarios. The proposed model only outperforms standard GNNs on the dataset introduced in this work.

---

### Official Review · Reviewer_AmiY · 2022-10-21

**Overall Score:** 6
**Confidence:** 4

**Review:**

Summary:
The authors perform an investigation into the choice of preprocessing of datasets which determines how the input feature matrix X of a graph is constructed. They find that how these initial embeddings are computed has a big effect on the performance of downstream machine learning models. They further find that the preprocessing of the data is one of the biggest factors in the overall performance, and has a bigger effect than using different architectures. Furthermore, they find that some unconnected deep learning methods can outperform graph based methods on some of these datasets. They introduce a new model GraNet intended to leverage the best of both types of models.

Strengths:
1. It is an interesting premise to consider the effect of different preprocessing methods affects GNN performance
2. A new model is introduced which has some promising performance based on preliminary experiments

Weaknesses:
1. Although many individual parts of the paper are well written, I think the overall narrative of the paper needs strengthening to make it more cohesive. I had to read it several times to properly connect the first part (assessing different embeddings) with the second part (introducing the new model); initially the two parts seemed a bit disconnected. After re-reading many times, I can now clearly connect the two sections, and have identified what I found confusing. In part 1, the other embedding methods appear to be much worse than Bag of Words in Table 2, so it made me wonder why they were considered. It’s true that you can see in which cases the model is able to compensate for the bad embeddings, but it felt a little irrelevant/contrived, given how much worse the performance was relative to Bag of Words (the original takeaway for me from the section, despite what the authors wrote, was that Bag of Words is *far* better than the alternatives). The embeddings chosen in Table 2 are different than those used in GraNet, which was then a bit confusing, and the claim in part 2 that using these different embeddings than BoW was superior felt a little murky given part 1. I think the writing can be clarified / improved to tie these sections together more clearly.
2. There are a lot of different things being introduced in these 4 pages; there is an exploration of an issue (effect of embedding choice), introduction of new datasets, and the introduction of a new model. I think it would have been better to narrow the focus a bit of the paper given the page limit, e.g. focusing on the effect of the choice of embedding, which would have enabled a more comprehensive view of it and would perhaps be more convincing and insightful to the reader. While the introduced model is interesting, I don’t feel it was fully tested/assessed given how few experiments were able to be done on it. (I’m aware that there are some additional tables in the appendix, but this is not ideal in my opinion as they are not based on something in the paper, and it is not immediately clear to the reader what the relevance of them are).

Questions:
1. What do you mean specifically by “large model”? (e.g. “mixing GNNs with large models”). Since this is a principal background aspect of the paper I think this should be defined more specifically somewhere.
2. Are the results listed the test results obtained from training done with a train/val/split, or are the test results from just a train/test split (it says it is a train/test split, but from the appendix it looks like a train/val/test split)?

I think the paper has some interesting insights to share with the community, and I think some rewriting to tie the two parts of the paper together will strengthen it. Overall I feel borderline about the paper because of my comment #2 in the weaknesses; there are a lot of components introduced and I wonder if an extended abstract is the correct format for introducing so many different things simultaneously. The format is short, so it does not feel like each aspect was presented thoroughly.

---

### Official Review · Reviewer_NM5u · 2022-10-23

**Overall Score:** 5
**Confidence:** 5

**Review:**

This paper tries to study the influence of initial input feature (embeddings) to the optimal GNN architecture. For example, Table2 shows that GAT2 favors the BOW feature, and GraphSAGE (Neighbor) favors Bytepair, roberta features, etc.

It is widely known that there's no a single GNN architecture that perform optimal over different types of graphs. However, most prior works focus on the structural perspective of graphs. For example, some GNN model favors homophilous graph [1], and there's also some works that combine multiple graph filter to get optimal solution [2].

[1] EvenNet: Ignoring Odd-Hop Neighbors Improves Robustness of Graph Neural Networks
[2] Demystifying graph neural network via graph filter assessment

This paper focuses on the input feature perspective, and the finding is also interesting. However, I'd like to see more analysis about why some GNN arcthiecture favors a particular input feature representation, and the ovservation is consistent under different hyperparameter setup (to make sure the conclusion is correct). Also, it would be good to include more GNN architectures instead of only GCN, GAT and GraphSAGE.


Also, I don't fully get what's the difference of the proposed GraNet against existing methods. It looks like using different input features fed to different (or same) GNN model plus a skip connection, which looks like an feature ensemble method. Please correct me if I understand wrong. In that case, if the Granet uses more features, it would be good to also compare with some naive baseline, such as concat all features together, and use a single MLP or GNN for prediction.

---

### Meta-Review · Area_Chair_kygn · 2022-11-19

**Confidence:** 4
**Recommendation:** Reject

**Meta Review:**

This paper studies the influence of input feature initializations on different GNN architectures. They find that these initial embeddings have a significant effect of the downstream performance on GNN models, even larger than the architecture of the GNN itself.

Reviewers agreed that the paper was tackling interesting questions and made for a promising submission, but that at this point, the paper needed another round of revisions to solidify its message and improve its presentation, as many part were unclear.

---

### Decision · Program_Chairs · 2022-11-23

Accept (Poster)